# You Only Cache Once:
# Decoder-Decoder Architectures for Language Models

**Yutao Sun**[*‡†]   **Li Dong**[*†]   **Yi Zhu**[†]   **Shaohan Huang**[†]
**Wenhui Wang**[†]   **Shuming Ma**[†]   **Quanlu Zhang**[†]   **Jianyong Wang**[‡]   **Furu Wei**[†◊]
[‡] Tsinghua University      [†] Microsoft Research
https://aka.ms/GeneralAI

## Abstract

We introduce a decoder-decoder architecture, YOCO, for large language models, which only caches key-value pairs once. It consists of two components, i.e., a *cross-decoder* stacked upon a *self-decoder*. The self-decoder efficiently encodes global key-value (KV) caches that are reused by the cross-decoder via cross-attention. The overall model behaves like a decoder-only Transformer, although YOCO only caches once. The design substantially reduces GPU memory demands, yet retains global attention capability. Additionally, the computation flow enables prefilling to early exit without changing the final output, thereby significantly speeding up the prefill stage. Experimental results demonstrate that YOCO achieves favorable performance compared to Transformer in various settings of scaling up model size and number of training tokens. We also extend YOCO to 1M context length with near-perfect needle retrieval accuracy. The profiling results show that YOCO improves inference memory, prefill latency, and throughput by orders of magnitude across context lengths and model sizes.

## 1   Introduction

Decoder-only Transformer [40] has become the de facto architecture for language models. By caching the previously computed key/value vectors, the model can reuse them for future generation steps. The key-value cache avoids encoding the history again for each token, greatly improving the inference speed. The compelling feature establishes the decoder-only architecture as the standard option.

However, as the number of serving tokens increases, the key-value (KV) caches occupy a lot of GPU memory, rendering the inference of large language models memory-bounded [29]. For the example of a 65B-size language model (augmented with grouped-query attention [1] and 8-bit KV quantization), 512K tokens occupy about 86GB GPU memory, which is even larger than the capacity of one H100-80GB GPU. In addition, the prefilling latency of long-sequence input is extremely high. For instance, using four H100 GPUs, the 7B language model (augmented with Flash-Decoding [6] and kernel fusion) requires about 110 seconds to prefill 450K tokens, and 380 seconds for 1M length. The above bottlenecks make it difficult to deploy long-context language models in practice.

In this work, we propose a decoder-decoder architecture, YOCO, for large language models, which only caches KV pairs once. Specifically, we stack cross-decoder upon self-decoder. Given an input sequence, the self-decoder utilizes efficient self-attention to obtain KV caches. Then the cross-decoder layers employ cross-attention to reuse the shared KV caches. The decoder-decoder architecture is conceptually similar to encoder-decoder, but the whole model behaves more like a decoder-only model from the external view. It naturally fits into autoregressive generation tasks, such as language

---

[*] Equal contribution. ◊ Corresponding author.

38th Conference on Neural Information Processing Systems (NeurIPS 2024).

modeling. First, because YOCO only caches once[2], the GPU memory consumption of KV caches is significantly reduced. Second, the computation flow of the decoder-decoder architecture enables prefilling to early exit before entering the self-decoder. The nice property speeds up the prefill stage dramatically, improving user experience for long-context language models. Third, YOCO allows for more efficient system design for distributed long-sequence training. In addition, we propose gated retention for self-decoder, which augments retention [35] with a data-controlled gating mechanism.

We conduct extensive experiments to show that YOCO achieves favorable language modeling performance and has many advantages in terms of inference efficiency. Experimental results demonstrate that YOCO can be scaled up with more training tokens, larger model size, and longer context length. Specifically, we scale up the 3B YOCO model to trillions of training tokens, attaining results on par with prominent Transformer language models, such as StableLM [39]. Moreover, the scaling curves ranging from 160M to 13B show that YOCO are competitive compared to Transformer. We also extend the context length of YOCO to 1M tokens, achieving near-perfect needle retrieval accuracy. In the multi-needle test, YOCO obtains competitive results even compared to larger Transformers.

In addition to good performance on various tasks, the profiling results show that YOCO improves the GPU memory footprint, prefill latency, throughput, and serving capacity. In particular, the memory of KV caches can be reduced by about $80\times$ for 65B models. Even for a 3B model, the overall inference memory consumption can be reduced by two times for 32K tokens and by more than nine times for 1M tokens. The prefill stage is speeded up by $71.8\times$ for the 1M context and $2.87\times$ for the 32K input. For example, for a 512K context, YOCO reduces the Transformer prefilling latency from 180 seconds to less than six seconds. The results position YOCO as a strong candidate model architecture for future large language models with native long-sequence support.

## 2 Related Work

Numerous efforts have been made to reduce KV caches for inference. Efficient attention mechanisms are proposed, such as sparse Transformer [4], linear attention [18], and recurrent modeling [27, 13, 46, 3, 19]. Another strand of research drops KV caches to achieve sparsity [49, 43, 11]. In comparison, we keep one global KV cache and still conduct full cross-attention for better long-context modeling. Moreover, some previous methods are complementary to our proposed architecture. For example, multi-/grouped-query attention [33, 1] and multi-latent attention [7] can be used in YOCO. Low-bit KV quantization [14, 25, 34] can also be used together to reduce memory consumption. In addition, the intriguing property of YOCO greatly speeds up the prefill stage.

## 3 You Only Cache Once (YOCO)

The proposed architecture, named YOCO, is designed for autoregressive modeling, such as large language models (LLMs). As shown in Figure 1, the decoder-decoder architecture has two parts, i.e., self-decoder and cross-decoder. Specifically, YOCO is stacked with $L$ blocks, where the first $\frac{L}{2}$ layers are self-decoder while the rest modules are cross-decoder. Given an input sequence $x = x_1 \cdots x_{|x|}$, the input embeddings are packed into $X^0 = [\boldsymbol{x}_1, \cdots, \boldsymbol{x}_{|x|}] \in \mathbb{R}^{|x| \times d_{\mathrm{model}}}$, where $d_{\mathrm{model}}$ is hidden dimension. We first obtain contextualized vector representations $X^l = \mathrm{Self\text{-}Decoder}(X^{l-1}), l \in [1, \frac{L}{2}]$, where $X^{L/2}$ is used to produce KV caches $\hat{K}, \hat{V}$ for cross-decoder. Then we compute $X^l = \mathrm{Cross\text{-}Decoder}(X^{l-1}, \hat{K}, \hat{V}), l \in [\frac{L}{2} + 1, L]$ to get the output vectors $X^L$. After obtaining $X^L$, a $\mathrm{softmax}$ classifier performs next-token prediction over the vocabulary.

Both self- and cross-decoder follow a similar block layout (i.e., interleaved attention and feed-forward network) as in Transformer [40]. We also include pre-RMSNorm [48], SwiGLU [32], and grouped-query attention [1] as improvements. The difference between the two parts lies in attention modules. Self-decoder uses efficient self-attention (e.g., sliding-window attention). In comparison, cross-decoder uses global cross-attention to attend to the shared KV caches produced by the output of the self-decoder. Notice that the whole model behaves like a decoder-only model. The tokens generated by cross-decoder are also fed back to self-decoder.

---

[2]The word "once" refers to global KV cache. Strictly, self-decoder also needs to store a certain number of caches. As the self-decoder utilizes an efficient attention module, the cache size is bounded to a constant, which can be ignored compared to global caches when the sequence length is large.

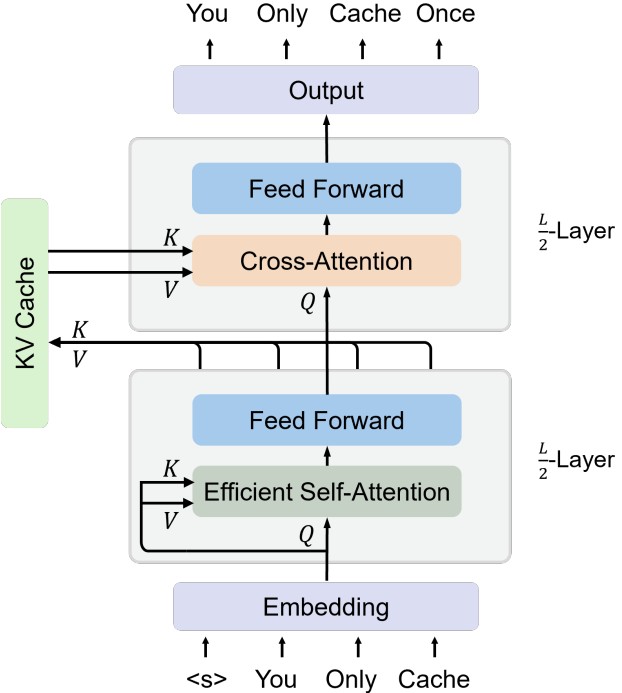

Figure 1: Overview of the decoder-decoder architecture. Self-decoder generates the global KV cache. Then cross-decoder employs cross-attention to reuse the shared KV caches. Both self-decoder and cross-decoder use causal masking. The overall architecture behaves like a decoder-only Transformer, autoregressively generating tokens.

## 3.1 Self-Decoder

Self-decoder takes embeddings $X^0$ as input and compute intermediate vector representation $X^{L/2}$:

$$Y^l = \text{ESA}(\text{LN}(X^l)) + X^l$$
$$X^{l+1} = \text{SwiGLU}(\text{LN}(Y^l)) + Y^l \tag{1}$$

where $\text{ESA}(\cdot)$ represents efficient self-attention, $\text{SwiGLU}(X) = (\text{swish}(XW_G) \odot XW_1)W_2$, and RMSNorm [48] is used for $\text{LN}(\cdot)$. Causal masking is used for efficient self-attention.

The key property of the efficient self-attention module is $\mathcal{O}(1)$ inference memory, i.e., constant number of KV caches. For example, the cache size of sliding-window attention [4] depends on the window size instead of the input length. More design choices (e.g., gated retention) of the efficient self-attention module are detailed in Section 4.

## 3.2 Cross-Decoder

First, the output of the self-decoder $X^{L/2}$ generates global KV caches $\hat{K}, \hat{V}$ for cross-decoder:

$$\hat{K} = \text{LN}(X^{L/2})W_K, \quad \hat{V} = \text{LN}(X^{L/2})W_V \tag{2}$$

where $W_K, W_V \in \mathbb{R}^{d \times d}$ are learnable. Then, cross-decoder layers are stacked after self-decoder to obtain the final output $X^L$. The KV caches $\hat{K}, \hat{V}$ are reused by all the $\frac{L}{2}$ cross-decoder modules:

$$Q^l = \text{LN}(X^l)W_Q^l$$
$$Y^l = \text{Attention}(Q^l, \hat{K}, \hat{V}) + X^l \tag{3}$$
$$X^{l+1} = \text{SwiGLU}(\text{LN}(Y^l)) + Y^l$$

where $\text{Attention}(\cdot)$ is standard multi-head attention [40], and $W_Q^l \in \mathbb{R}^{d \times d}$ is a learnable matrix. Causal masking is also used for cross-attention. Because cross-attention is compatible with group query attention [1], we can further save the memory consumption of KV caches.

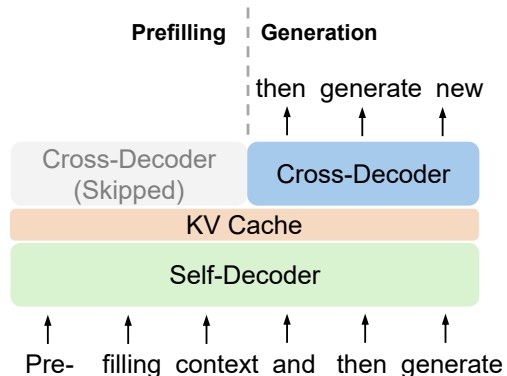

Figure 2: YOCO Inference. **Prefill**: encode input tokens in parallel. **Generation**: decode output tokens one by one. The computation flow enables prefilling to early exit without changing the final output, thereby significantly speeding up the prefill stage.

| | KV Cache Memory |
|---|---|
| Transformer | $\mathcal{O}(LND)$ |
| YOCO | $\mathcal{O}((N+L)D)$ |

Table 1: Inference memory complexity of KV caches. $N, L, D$ are the sequence length, number of layers, and hidden dimension.

| | Prefilling Time |
|---|---|
| Transformer | $\mathcal{O}(LN^2D)$ |
| YOCO | $\mathcal{O}(LND)$ |

Table 2: Prefilling time complexity of attention modules. $N, L, D$ are the same as above.

## 3.3 Inference Advantages

In addition to competitive language modeling results, YOCO significantly reduces serving costs and improves inference performance. We report detailed inference comparisons in Section 5.4.

**Saving GPU Memory and Serving More Tokens.** Table 1 compares the memory complexity between Transformers and YOCO. Specifically, because global KV caches are reused and efficient self-attention needs constant caches, the number of caches is $\mathcal{O}(N + CL)$, where $N$ is the input length, $C$ is a constant (e.g., sliding window size), and $L$ is the number of layers. For long sequences, $CL$ is much smaller than $N$, so about $\mathcal{O}(N)$ caches are required, i.e., you only cache once. In comparison, Transformer decoders have to store $N \times L$ keys and values during inference. So YOCO roughly saves $L$ times GPU memory for caches compared to Transformer. Because the bottleneck of inference capacity becomes KV caches, our method enables us to serve many more tokens without being out of GPU memory. The increased batch size is also beneficial to inference throughput.

**Reducing Prefilling Time and Improving Throughput.** As shown in Figure 2, because the cross-decoder reuses the outputs of self-decoder, we can exit early before entering the cross-decoder during the prefill stage. The intriguing property of computation dependency greatly accelerates the prefilling speed. First, only half the layers are needed for forward computation, i.e., at least half prefilling latency reduction. Second, the efficient attention modules of the self-decoder are usually fast. For the example of 512K context length, we can decrease the prefilling latency from 180 seconds (Transformer with optimized inference, such as Flash-Decoding) to less than 6 seconds (Figure 9). Even for 32K length, YOCO has about three times speedup in terms of prefilling time. Table 2 compares prefilling time complexity of attention modules between Transformer and YOCO.

## 4 Design Choices of Self-Decoder

We can choose various efficient self-attention methods for self-decoder. As long as the module only requires constant inference memory, the self-decoder's cache memory complexity depends on the number of layers. Moreover, a good module choice improves both training and deployment costs. In this work, we use sliding-window attention (Section 4.1) or gated retention (Section 4.2).

## 4.1 Sliding-Window Attention

Sliding-window attention [4] restricts the attention range into a fixed window size $C$. In contrast, vanilla Transformer decoders attend to all previous tokens. During inference, the KV cache memory complexity can be reduced from $\mathcal{O}(N)$ to $\mathcal{O}(C)$, i.e., the memory usage is constant rather than increasing with sequence length. Similar to multi-head self-attention [40], we compute the output of

sliding-window attention via:

$$Q = XW_Q, \quad K = XW_K, \quad V = XW_V$$

$$\text{head}_i = \text{softmax}(Q_{[i]}K_{[i]}^{\mathsf{T}} + B)V, \quad B_{ij} = \begin{cases} 0, & i - C < j \leq i \\ -\infty, & \text{otherwise} \end{cases} \tag{4}$$

$$\text{SWA}(X) = \text{Concat}(\text{head}_1, \cdots, \text{head}_h)W_O$$

where $W_Q, W_K, W_V, W_O \in \mathbb{R}^{d \times d}$ are learnable matrices, and the window causal mask $B$ controls each query only attends to the previous keys whose distances are less than $C$. The pre-normalization and residual connection are also applied to the module.

## 4.2 Gated Retention

Gated retention (gRet, aka gRetNet) augments retention [35] with a data-dependent gating mechanism. We use gRet as the default efficient self-attention module. The method unifies the parallel, recurrent, and chunkwise recurrent computation paradigms, which are equivalent and can obtain the same computation results. The training process usually uses the parallel or chunkwise recurrent paradigms, while the inference stage can employ the recurrent paradigm for constant KV memory.

**The Parallel Representation** The gated retention is defined as:

$$Q = (XW_Q) \odot \Theta, \quad K = (XW_K) \odot \overline{\Theta}, \quad V = XW_V, \quad \Theta_n = e^{in\theta}$$

$$\gamma = \text{sigmoid}(XW_\gamma)^{1/\tau}, \quad D_{nm} = \begin{cases} \prod_{i=m+1}^{n} \gamma_i, & n \geq m \\ 0, & n < m \end{cases} \tag{5}$$

$$\text{gRet}(X) = (QK^{\mathsf{T}} \odot D)V$$

where $W_Q, W_K, W_V \in \mathbb{R}^{d \times d}$ and $W_\gamma \in \mathbb{R}^{d \times 1}$ are learnable weights, and the temperature term $\tau$ encourages $\gamma$ to 1 for better memorization [46]. The data-controlled decay is head-wise [19] rather than element-wise so that the computation can fully utilize NVIDIA tensor cores. Refer to [35] for more details about the other designs.

**The Recurrent Representation** Being equivalent to Equation (5), the output of gated retention can be computed recurrently. For the $n$-th timestep, the output is obtained via:

$$S_n = \gamma_n S_{n-1} + K_n^{\mathsf{T}} V_n$$
$$\text{gRet}(X_n) = Q_n S_n, \quad n = 1, \cdots, |x| \tag{6}$$

where $Q, K, V, \gamma$ are the same as in Equation (5). During auto-regressive inference, the self-decoder maintains $S_n$ as the intermediate state for an efficient generation.

**The Chunkwise Recurrent Representation** The chunk-wise representation is a unified formulation of recurrent and parallel representations. Given chunk size $B$, the outputs are computed chunk by chunk. The computation is divided into inner-chunk and cross-chunk parts. Denote $[i]$ as the $i$-th chunk, i.e., $x_{[i]} = x_{(i-1)B+1}, \cdots, x_{iB}$, we compute the $i$-th chunk as:

$$\beta_{(i-1)B+j} = \prod_{k=(i-1)B+1}^{(i-1)B+j} \gamma_k, \quad D_{[i]}(j,k) = \frac{\beta_{(i-1)B+k}}{\beta_{(i-1)B+j}} \text{ if } j \leq k \text{ else } 0$$

$$R_i = K_{[i]}^{\mathsf{T}}(V_{[i]} \odot \frac{\beta_{iB}}{\beta_{[i]}}) + \beta_{iB} R_{i-1}, \quad \beta_{[i]}(j,k) = \beta_{(i-1)B+j} \tag{7}$$

$$\text{gRet}(X) = \underbrace{(Q_{[i]}K_{[i]}^{\mathsf{T}} \odot D_{[i]})V_{[i]}}_{\text{Inner-Chunk}} + \underbrace{(Q_{[i]}R_{i-1}) \odot \beta_{[i]}}_{\text{Cross-Chunk}}$$

where $R_i$ is the intermediate state of the $i$-th chunk, and $\beta$ summarizes the data-controlled decay $\gamma$. Appendix B proves the equivalence between the computation paradigms. The chunkwise paradigm combines the best of parallelism and recurrence, i.e., saving FLOPs compared to fully parallel computation and reducing iterations compared to recurrent computation. During the training and prefill stages, the chunk-wise representation increases throughput and reduces GPU memory consumption.

| Model | ARC-C | ARC-E | BoolQ | Hellaswag | OBQA | PIQA | Winogrande | SciQ | Avg |
|---|---|---|---|---|---|---|---|---|---|
| *Training with 1T tokens* | | | | | | | | | |
| OpenLLaMA-3B-v2 [12] | 0.339 | 0.676 | **0.657** | **0.700** | 0.260 | 0.767 | 0.629 | **0.924** | 0.619 |
| StableLM-alpha-3B-v2 [38] | 0.324 | 0.673 | 0.646 | 0.686 | 0.264 | 0.760 | 0.621 | 0.921 | 0.612 |
| StableLM-3B-4E1T [39] | — | 0.666 | — | — | — | **0.768** | 0.632 | 0.914 | — |
| YOCO-3B | **0.379** | **0.731** | 0.645 | 0.689 | **0.298** | 0.763 | **0.639** | **0.924** | **0.634** |
| *Training with 1.6T tokens* | | | | | | | | | |
| StableLM-3B-4E1T [39] | — | 0.688 | — | — | — | 0.762 | 0.627 | 0.913 | — |
| YOCO-3B | 0.396 | 0.733 | 0.644 | 0.698 | 0.300 | 0.764 | 0.631 | 0.921 | 0.636 |
| *Extending context length to 1M tokens* | | | | | | | | | |
| YOCO-3B-1M | 0.413 | 0.747 | 0.638 | 0.705 | 0.300 | 0.773 | 0.651 | 0.932 | 0.645 |

Table 3: Eval Harness [10] accuracy compared with well-trained Transformer language models. We scale the 3B model to 1.6 trillion training tokens. The 1T and 1.6T results of StableLM-3B-4E1T are taken from its technical report [39]. YOCO-3B-1M is extended to the context length of 1M tokens.

**Multi-Head Gated Retention** Similar to multi-head attention [40] and multi-scale retention [35], we apply gated retention to each head and combine the outputs together:

$$
\begin{aligned}
\text{head}_i &= \text{gRet}(X) \\
Y &= \text{GroupNorm}_h(\text{Concat}(\text{head}_1, \cdots, \text{head}_n)) \\
\text{MHGR}(X) &= (\text{swish}(XW_G) \odot Y)W_O
\end{aligned}
\tag{8}
$$

where $W_G, W_O \in \mathbb{R}^{d \times d}$ are learnable matrices, and $\text{GroupNorm}$ [42] normalizes each head [41]. We also apply swish gate to increase non-linearity [35].

## 5 Experiments

We evaluate YOCO for large language models from the following perspectives. First, we follow the setting of StableLM-3B-4E1T [39] to scale up training tokens (Section 5.1). Second, we present the scaling curves of the proposed architectures (Section 5.2). Third, we scale up the YOCO model to 1M context length and evaluate its long-sequence modeling capability (Section 5.3). Fourth, we analyze the deployment advantages, including GPU memory footprint, serving capacity, prefilling time, and throughput (Section 5.4). Experimental results show that YOCO achieves competitive performance in various evaluation metrics and significantly reduces the inference cost.

### 5.1 Language Modeling Evaluation

We train a 3B-size YOCO language model by scaling up the number of training tokens. Then we compare the checkpoints with strong Transformer-based language models. We use a similar training recipe to that in StableLM-3B-4E1T [39]. Detailed hyperparameters are described in Appendix D.

**Results** Table 3 compares YOCO with OpenLLaMA-v2-3B [12], StableLM-base-alpha-3B-v2 [38], and StableLM-3B-4E1T [39]. We use LM Eval Harness [10] to evaluate zero-shot performance on various downstream tasks. OpenLLaMA-v2-3B and StableLM-base-alpha-3B-v2 are trained with 1T tokens. The intermediate numbers of StableLM-3B-4E1T are taken from its technical report [39]. Experimental results indicate that YOCO achieves comparable results with previous well-tuned Transformer language models. Both the checkpoints trained with 1T tokens and 1.6T tokens obtain a consistent trend. Moreover, the results show that YOCO is scalable in terms of training tokens.

### 5.2 Scalability Compared with Transformers

We compare the scaling curves between Llama Transformer [40, 37], YOCO with gated retention (YOCO$_{\text{gRet}}$; Section 4.2), and YOCO with sliding-window attention (YOCO$_{\text{SWA}}$; Section 4.1). We train language models of various sizes (i.e., 160M, 400M, 830M, 1.4B, 2.7B, 6.8B, and 13B) using the same training data and settings. We augment the Transformer architecture with Llama [37] improvements, such as RMSNorm [48], SwiGLU [32], and removing bias. The sliding window size of YOCO$_{\text{SWA}}$ is 1,024. The training batch size is 0.25M tokens with a 2k sequence length. We train the models with 40k steps, i.e., 10B tokens. In practice, we find that the setting is effective for loss convergence, and scaling laws can be well-fitted. More hyperparameters are detailed in Appendix E.

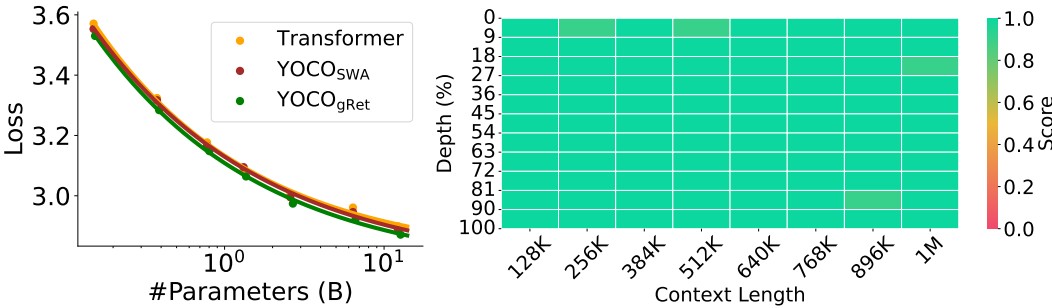

Figure 3: LM loss decreases along with scaling up the model size (ranging from 160M to 13B).

Figure 4: Needle-in-a-haystack results in 1M length.

**Results** Figure 3 reports the validation loss with various parameter counts. We also fit the scaling curves as in [17]. YOCO obtains comparable performance from 160M to 13B compared to the Llama-optimized transformer architecture. The findings demonstrate that YOCO scales effectively with respect to model size. Moreover, YOCO$_{gRet}$ outperforms Transformer and YOCO$_{SWA}$. The gains come from hybrid architectures of attention and retention, whose inductive biases tend to be complementary to each other. Recent hybrid architectures [21] also confirm similar findings.

### 5.3 Long-Context Evaluation

We extend the context length of YOCO-3B (Section 5.1) to 1M tokens. We continue the model training with longer lengths progressively. The length schedule is 64K, 256K, and 1M tokens. Training data is up-sampled according to sequence length [9]. For a fair comparison, we do not use long-instruction tuning data. More training details are described in Appendix F.

**Needle In A Haystack with 1M Context** The pressure test evaluates whether models can retrieve "needles" from a long document [16]. We follow the evaluation setting of Gemini 1.5 [30] and LWM [24]. The needles are constructed as a city with a magic number. We run 10 times at the same depth and length. The average accuracy is reported. Figure 4 shows that YOCO-3B-1M passes the Needle-In-A-Haystack test with near perfect accuracy. The results indicate that YOCO has strong long-context modeling capability.

**Multi-Needle Retrieval** Besides single-needle retrieval, we conduct a multi-needle evaluation. We compare YOCO-3B-1M with previous long-context language models, including MiniCPM-128K [15], ChatGLM3-128K [47], YaRN-Mistral-128K [28], and LWM-1M-text [24]. The evaluation is conducted in 128K sequence length, because most previous models are tuned with this length.

Table 4 presents accuracy results with $N$ needles. LWM-1M-text and YOCO-3B-1M are trained with a 1M context length, while the others are of 128K length. Although LWM-1M-text continues training of Llama-2-7B, YOCO-3B-1M can still achieve comparable performance with half the model size. Moreover, the 7B-size YaRN-Mistral-128K [28] obtained by position interpolation lags behind the other models. Compared to MiniCPM-128K and ChatGLM3-128K, YOCO-3B-1M also outperforms these well-trained language models.

| Model | Size | $N = 1$ | $N = 2$ | $N = 4$ | $N = 8$ |
|---|---|---|---|---|---|
| YaRN-Mistral-128K [28] | 7B | 0.02 | 0.12 | 0.08 | 0.20 |
| LWM-1M-text [24] | 7B | 1.00 | 0.90 | 0.76 | 0.62 |
| MiniCPM-128K [15] | 2.4B | 1.00 | 1.00 | 0.54 | 0.56 |
| ChatGLM3-128K [47] | 6B | 0.94 | 0.72 | 0.52 | 0.44 |
| YOCO-3B-1M | 3B | 0.98 | 0.98 | 0.84 | 0.56 |

Table 4: Multi-needle retrieval accuracy. $N$ indicates the number of needles. $N = 1$ is single-needle retrieval used as a reference, and $N > 1$ indicates the multi-needle test. The evaluation is conducted in 128K length, because most previous long-context models are tuned with this length.

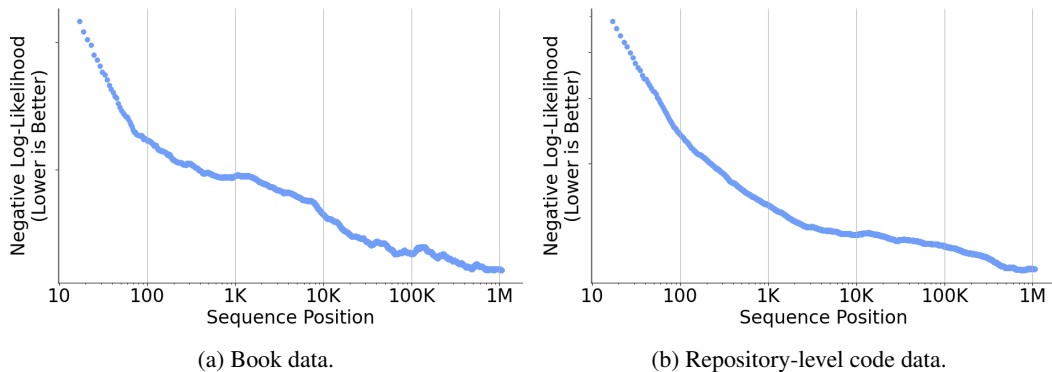

| (a) Book data. | (b) Repository-level code data. |

Figure 5: Cumulative average negative log-likelihood on book and repository-level code. We filter the validation examples that are longer than 1M tokens. YOCO achieves improved performance with longer context, i.e., utilizing long-distance information for language modeling.

**Perplexity over Long Sequences**   Figure 5 shows the cumulative average negative log-likelihood (NLL) as a function of context length. We evaluate both book and repository-level code data. We follow the setting of [30] and filter validation data that are longer than 1M tokens. NLL decreases consistently with longer sequence length. The results indicate that YOCO can effectively utilize long-distance dependency for language modeling. We also observe that the NLL-length curves tend to fit the power law, where the gaps are affected by the noise within the validation examples.

## 5.4   Inference Advantages

We analyze inference efficiency from various perspectives, such as GPU memory footprint, prefilling latency, throughput, and serving capacity. We show that YOCO reduces the deployment cost by orders of magnitude, especially for long-sequence inference. More importantly, the user experience (such as latency) is improved while maintaining good performance and reducing expenses.

We compare $YOCO_{gRet}$ with Transformer. The default model configuration follows Section 5.1. Notice that Transformer uses grouped-query attention [1], Flash-Decoding [6], and kernel fusion for a fair comparison. As described in Section 4.2, gated retention uses the chunk-recurrent representation in the prefill stage, and the recurrent representation in the generation stage. The chunk size is set to 256. We implement a Triton [36] kernel for gated retention. The evaluation sequence length ranges from 32K to 1M. The last 1,024 tokens are supposed to be generated, while the previous tokens are given input context. The experiments are conducted with H100-80GB GPU cards.

**GPU Memory**   The inference memory consumption is made up of three parts, namely model weights, intermediate activation, and KV cache. Figure 6 presents the breakdown memory profiling results. Along with an increase in context length, the main memory bottleneck becomes KV caches, while model weights consume constant memory. The results show that $YOCO_{gRet}$ alleviates the activation cost and KV cache memory footprint.

As shown in Figure 7, the memory cost is significantly reduced using YOCO. Moreover, the memory consumption of YOCO increases slowly along the sequence length. For example of 1M length, the overall inference memory usage is only 12.4GB, while Transformers occupy $9.4\times$ GPU memory. YOCO makes it feasible to deploy long-sequence modeling on customer-level GPUs. Even with a 32K sequence length, YOCO requires about

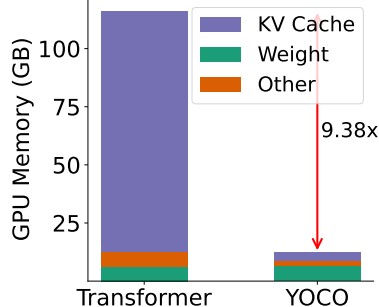

Figure 6: Breakdown memory consumption in 1M context length.

$2\times$ less memory than Transformer. Although we compare 3B-size models here, the reduction ratio becomes larger as the number of layers increases.

Figure 8 reports the GPU memory consumption of KV cache for each token. As YOCO only caches one layer of global key-value pairs, it needs roughly $L$ times less memory compared to Transformer.

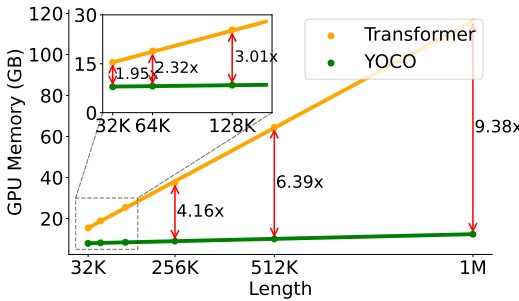

Figure 7: Inference memory of Transformer and YOCO across various lengths.

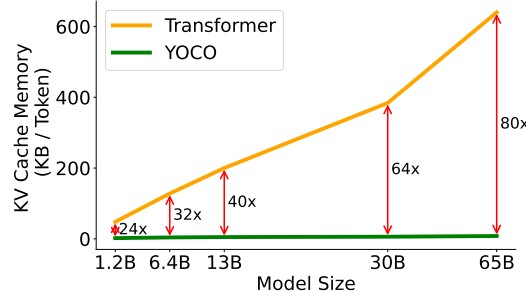

Figure 8: GPU memory of KV cache for each token with different model size.

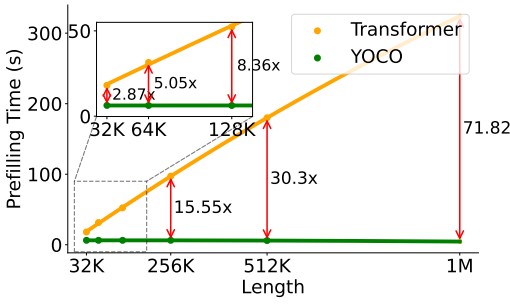

Figure 9: Prefilling latency for different lengths. Transformer's time grows quadratically while YOCO's grows linearly.

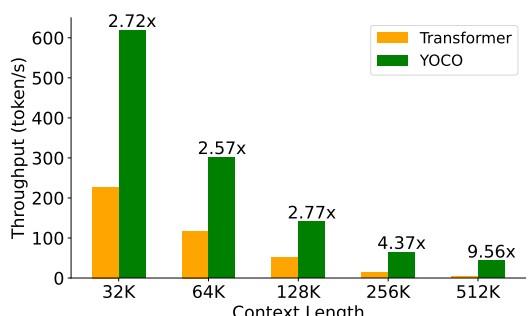

Figure 10: Inference throughput of Transformer and YOCO varying the context length.

For example, YOCO can serve 128K tokens with 1GB GPU memory, while Transformer with GQA [1] can only support 1.6K tokens at 65B model size.

**Prefilling Latency** In the prefill stage, the model encodes input tokens in parallel. As shown in Figure 9, the prefilling latency is a pain point of user experience for long-context models. For 512K- and 1M-length input sequences, Transformer needs about 180 seconds and 300 seconds, respectively. The computational complexity of Transformer is $\mathcal{O}(N^2)$, which requires a large number of FLOPs for long context. In contrast, YOCO's prefilling time is $\mathcal{O}(N)$, growing linearly (Section 3.3) along the sequence length. Figure 9 shows that YOCO reduces the Transformer prefilling time from 180 seconds to less than 6 seconds for 512K context. As described in Section 3.3, the prefill stage can early exit before entering cross-decoder. So, there is at least two times speedup of prefilling latency even for short context. For example, YOCO is $2.87\times$ faster than Transformer for 32K length.

**Throughput** The throughput indicates how many tokens the model can process per second, involving both pre-filling and generation time. Figure 10 shows that YOCO achieves higher throughput across context lengths compared to Transformer. For the example of 512K queries, Transformer's throughput is 4.5 token/s while YOCO reaches 43.1 token/s, i.e., achieving $9.6\times$ speedup. The throughput is improved for the following reasons. First, YOCO decreases the time required for prefilling as previously demonstrated. Second, as the memory consumption is reduced, we can use larger batch size for inference, which also contributes to the throughput improvement.

## 5.5 Comparisons with Transformer Variants

We compare YOCO$_{\text{gRet}}$ and YOCO$_{\text{SWA}}$ with Transformer and other variants, including H3 [5], RetNet [35], Mamba [13], and gRetNet (Section 4.2). All models have 160M parameters with 12 layers and a hidden dimension of 768. The weights of word embedding and $\mathrm{softmax}$ projection are shared. For Mamba, we follow the details in [13], where double-SSM layers are implemented instead of "SSM + SwiGLU". For H3, the experiment uses a hybrid version following the original paper [5], where two attention layers are inserted after the first and $\frac{L}{2}$-th layers.

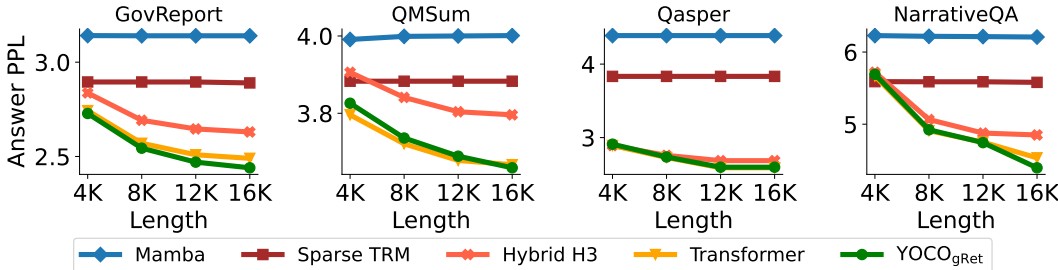

Figure 11: Long sequence task perplexity decreases along with the increasing input length.

**Fine-Grained LM Perplexity** Table 5 reports the fine-grained validation perplexity for language modeling. Following Zoology [2], we divide the perplexity into "*Ar-Hit*" and "*First-Occur*". Specifically, "*Ar-Hit*" considers the predicted tokens that are bigrams previously seen in the previous context, which evaluates the associative recall capability. "*First-Occur*" considers the tokens that cannot be recalled from the context.

|  | Valid. Set↓ | AR-Hit↓ | First-Occur↓ |
|---|---|---|---|
| Mamba [13] | 3.645 | 1.555 | 4.126 |
| RetNet [35] | 3.633 | 1.466 | 4.131 |
| Hybrid H3 [5] | 3.591 | 1.251 | 4.130 |
| gRetNet | 3.600 | 1.354 | 4.116 |
| Transformer | 3.564 | 1.219 | 4.104 |
| YOCO$_{\text{SWA}}$ | 3.553 | 1.202 | 4.094 |
| YOCO$_{\text{gRet}}$ | **3.530** | **1.199** | **4.067** |

Table 5: Fine-grained LM perplexity results.

**Long-Context Evaluation** Figure 11 reports the answer perplexity with varying context length (ranging from 4K to 16K) on the ZeroSCROLLS [31] benchmark. We continue training the above models in 16,384 length with 2B tokens. The rotation base scaling [44] is used for length extension. For sparse Transformer, we use the context window of 2,048 and keep RoPE $\theta$ unmodified. As shown in Figure 11, YOCO and Transformer consistently outperform other methods across tasks and lengths, which is consistent with the findings in Section 5.3. Moreover, the results highlight the importance of global attention for long-context modeling. Notice that the 12K and 16K results in Qasper are similar because the lengths of most documents are shorter than 16K.

### 5.6 Ablation Studies

As shown in Table 6, we explore different layout configurations for YOCO. First, we compare the ratio of self-decoder to cross-decoder layers. For example, YOCO$_{[1:1]}$ is the default setting, where each module contains $L/2$ layers. The results show that YOCO$_{[1:1]}$ is comparable to YOCO$_{[3:1]}$ and outperforms both YOCO$_{[1:3]}$ and YOCO$_{[0:1]}$. We use [1:1] as the default layout. Future work can refine a scaling law to guide the choice of

|  | Valid. Set↓ | AR-Hit↓ | First-Occur↓ |
|---|---|---|---|
| YOCO$_{[1:1]}$ | 3.530 | 1.199 | 4.067 |
| YOCO$_{[3:1]}$ | 3.526 | 1.207 | 4.060 |
| YOCO$_{[1:3]}$ | 3.565 | 1.230 | 4.102 |
| YOCO$_{[0:1]}$ | 3.898 | 1.827 | 4.374 |
| Unstacked YOCO$_{[1:1]}$ | 3.531 | 1.188 | 4.071 |
| Interleaved & Hybrid | 3.542 | 1.204 | 4.081 |

Table 6: Fine-grained LM perplexity results. "$[s{:}c]$" is the ratio of self-decoder to cross-decoder layers.

layer ratio. Second, the setting "Unstacked YOCO$_{[1:1]}$" uses word embeddings $X^0$ as input to the cross-decoder, rather than stacking cross-decoder upon self-decoder (i.e., using $X^{L/2}$ in Equation (3)). Third, the model "Interleaved & Hybrid" is a hybrid architecture that interleaves gRetNet and Transformer layers.

## 6  Conclusion

In this work, we propose a decoder-decoder architecture (YOCO) for large language modeling. YOCO achieves significantly better inference efficiency and competitive performance compared to Transformers. Experimental results demonstrate that YOCO achieves favorable results for large language models in various settings, i.e., scaling up the number of training tokens, scaling up model size, and scaling up context length to 1M tokens. Profiling results also show that YOCO improves inference efficiency by orders of magnitude, especially for long-sequence modeling.

## Acknowledgement

We would like to acknowledge Ben Huntley for maintaining the GPU cluster. The long-sequence training utilizes CUBE, which is an internal version of [22]. We implement the Triton kernel of gated retention based on FLA [45]. This work was supported in part by National Key Research and Development Program of China under Grant No. 2020YFA0804503, National Natural Science Foundation of China under Grant No. 62272264, and Beijing Academy of Artificial Intelligence.

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

## A    Chunk Parallelism for Long-Sequence Training of YOCO

We introduce chunk parallelism for YOCO to reduce the communication frequency, accelerating long-sequence training in Section 5.3. Dividing long sequences into different devices is essential when the training length is extremely long [20, 8]. However, the overall throughput tends to be bounded by GPU communication [23]. Cross-decoder disentangles self-attention dependency while preserving modeling capability, bringing intriguing advantages to distributed long-sequence training.

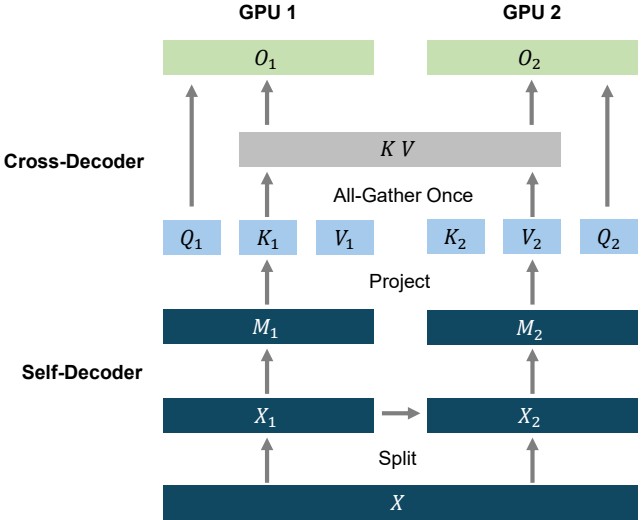

Figure 12: Chunk parallelism of YOCO training on two GPU devices. The training strategy is to partition the sequence into different chunks. $M$ denotes the intermediate representation $X^{L/2}$, i.e., the output of self-decoder. The keys and values in the cross-decoder are only gathered once.

In self-decoder, the dependency only exists in the adjacent devices. For example, gated retention only requires the hidden state $S_n$ in Equation (6), and sliding-window attention attends to tokens within the context window. Therefore, the communication amount of self-decoder is relatively small. In the cross-decoder, the all-gather operation is only triggered once for the KV cache, rather than communicating in each layer. The hardware-friendly architecture gives more flexibility to distributed long-sequence training.

## B    Chunk-wise Representation of Gated Retention

We illustrate the equivalence between recurrent representation and chunkwise recurrent representation of gated retention. For the output $O_n$, $n$ can be split as $n = kB + r$ where $B$ is the chunk size:

$$O_n = \sum_{m=1}^{n} \prod_{i=m+1}^{n} \gamma_i Q_n K_m^{\mathsf{T}} V_m = \sum_{m=kB+1}^{n} \prod_{i=m+1}^{n} \gamma_i Q_n K_m^{\mathsf{T}} V_m + \sum_{m=1}^{kB} \prod_{i=m+1}^{n} \gamma_i Q_n K_m^{\mathsf{T}} V_m$$

$$\sum_{m=kB+1}^{n} \prod_{i=m+1}^{n} \gamma_i Q_n K_m^{\mathsf{T}} V_m = (Q_n K_{kB+1:n}^{\mathsf{T}} \odot \Gamma_{kB+1:n}) V_{kB+1:n}$$

$$\sum_{m=1}^{kB} \prod_{i=m+1}^{n} \gamma_i Q_n K_m^{\mathsf{T}} V_m = (Q_n \prod_{i=kB+1}^{n} \gamma_i) \sum_{c=0}^{k-1} \sum_{m=1}^{B} (K_{m+cB}^{\mathsf{T}} V_{m+cB} \prod_{i=m+cB+1}^{(c+1)B} \gamma_i) \prod_{i=(c+1)B+1}^{kB} \gamma_i$$

$$= (Q_n \prod_{i=kB+1}^{n-1} \gamma_i) \sum_{c=1}^{k} (K_{[c]}^{\mathsf{T}} (V_{[c]} \odot \zeta_{[c]})) \prod_{i=c+1}^{k} \alpha_i$$

$$= (Q_n \prod_{i=kB+1}^{n-1} \gamma_i) R_{i-1}$$

$$(9)$$

where $\Gamma_i = \prod_{k=i+1}^{n} \gamma_i$, $\zeta_{[c]}(j,k) = \prod_{i=(c-1)B+j+1}^{cB} \gamma_i$, $\alpha_i = \prod_{j=(i-1)B+1}^{iB} \gamma_j$, $[i]$ indicates the $i$-th chunk, i.e., $x_{[i]} = [x_{(i-1)B+1}, \cdots, x_{iB}]$. $R_n$ is written as a recurrent function:

$$R_i = K_{[i]}^{\mathsf{T}}(V_{[i]} \odot \zeta_{[i]}) + \alpha_i R_{i-1} \tag{10}$$

Denote $[i]$ as the $i$-th chunk, i.e., $x_{[i]} = [x_{(i-1)B+1}, \cdots, x_{iB}]$, $\beta_{(i-1)B+j} = \prod_{k=(i-1)B+1}^{(i-1)B+j}$, $\beta_{[i]}(j,k) = \beta_{(i-1)B+j}$, We concatenate the output in a block together:

$$O_{[n]} = \sum_{m=kB+1}^{[n]} \beta_{[n]} Q_{[n]} K_m^{\mathsf{T}} V_m + \sum_{m=1}^{kB} \beta_{[n]} Q_{[n]} \prod_{i=m+1}^{n} \gamma_i K_m^{\mathsf{T}} V_m$$

$$\sum_{m=kB+1}^{[n]} \beta_{[n]} Q_{[n]} K_m^{\mathsf{T}} V_m = (Q_{[n]} K_{[n]}^{\mathsf{T}} \odot D_{[n]}) V_{[n]}, \quad D_{[n]}(j,k) = \frac{\beta_{(n-1)B+k}}{\beta_{(n-1)B+j}} \text{ if } j \le k \text{ else } 0$$

$$\sum_{m=1}^{kB} \beta_{[n]} Q_{[n]} \prod_{i=m+1}^{n} \gamma_i K_m^{\mathsf{T}} V_m = \beta_{[n]} Q_{[n]} R_{i-1}, \quad R_i = K_{[i]}^{\mathsf{T}}(V_{[i]} \odot \frac{\beta_{iB}}{\beta_{[i]}}) + \beta_{iB} R_{i-1},$$

$$O_{[n]} = \underbrace{(Q_{[n]} K_{[n]}^{\mathsf{T}} \odot D_{[n]}) V_{[n]}}_{\text{Inner-Chunk}} + \underbrace{(Q_{[n]} R_{n-1}) \odot \beta_{[n]}}_{\text{Cross-Chunk}}$$

$$\tag{11}$$

Finally, we show that the chunkwise recurrent representation of gated retention is equivalent to the other two representations.

## C   Pseudo Code of Gated Retention

We present pseudocode for the three computation paradigms of gated retention (Section 4.2). Parallel implementation enables training parallelism to fully utilize GPUs. The recurrent paradigm enables low-cost inference. Chunkwise retention combines the above advantages (i.e., parallel within each chunk and recurrent across chunks), which has linear memory complexity for long sequences.

```
def ParallelRetention(
    q, # bsz * num_head * len * dim
    k, # bsz * num_head * len * dim
    v, # bsz * num_head * len * dim
    gt): # bsz * num_head * len
    retention = q @ k.transpose(-1, -2)
    causal_mask = torch.full([q.shape[-2], q.shape[-2]], float("-inf"), device=q.device).
        triu(1).type_as(q)
    gt = F.logsigmoid(gt).cumsum(-1) / gate_logit_normalizer
    mask = (g[..., None] - g[..., None, :] + causal_mask).exp()

    retention = retention * mask
    output = retention @ v
    output = group_norm(output)
    return output
```

```
def RecurrentRetention(
    q, k, v, # bsz * num_head * dim
    past_kv, # bsz * num_head * dim * dim
    gt # bsz * num_head * 1 * 1
    ):
    gt = F.logsigmoid(gt) / gate_logit_normalizer
    current_kv = gt.exp() * past_kv + k.unsqueeze(-1) * v.unsqueeze(-2)
    output = torch.sum(q.unsqueeze(-1) * current_kv, dim=-2)
    output = group_norm(output)
    return output, current_kv
```

```
def ChunkwiseRetention(
    q, k, v, # bsz * num_head * chunk_size * dim
    past_kv, # bsz * num_head * dim * dim
    gt): # bsz * num_head * chunk_size
    gt = F.logsigmoid(gt).cumsum(-1) / gate_logit_normalizer
    cross_retention = (q @ past_kv) * gt[..., None].exp()
    inner_retention = ParallelRetention(q, k, v, gt)
    retention = inner_retention + cross_retention
    output = group_norm(retention)

    value_decay = (-gt + gt[:, :, :, -1, None]).exp()[..., None]
    chunk_decay = gt[..., -1].exp()
    current_kv = chunk_decay * past_kv + k.transpose(-1, -2) @ (v * value_decay)
    return output, current_kv
```

## D   Hyperparameters for YOCO-3B (Section 5.1)

We adjust the head dimension to 128 instead of 80 as in StableLM for better kernel support. To keep the model size unchanged, we set the hidden size to 3072 and the number of layers to 26. Grouped-query attention [1] is used, where the number of query heads is 24, and the number of key-value heads is 8. We train YOCO with gated retention (Section 4.2). The non-embedding parameter count is 2.8B. In comparison, StableLM-3B-4E1T is 2.7B and OpenLLaMA-v2-3B [12] is 3.2B. The training sequence length is 4096. The batch size is 4M tokens. We use the AdamW [26] optimizer with $\beta = 0.9, 0.95$. The maximal learning rate is 3.2e-4 with 1000 warmup steps and linear decay to 1.28e-5. The total schedule is set to 5T tokens. Given the resource budget, we train the model with 400k steps (1.6T tokens). The curated training corpus is similar to [39]. We use `tiktoken-cl100k_base` as the tokenizer. The hidden dimension is set to 3072. The number of layers is 26. The number of query heads is 24, and the number of key/value heads is 8 with grouped-query attention [1]. The total number of parameters without embedding is 2.83B. The training batch size is 4M tokens. We use 4096 training length. The optimizer is AdamW [26] with $\beta = (0.9, 0.95)$. The learning rate is $3.2 \times 10^{-4}$ with 1000 warmup steps. We set a 5T-token learning rate schedule with linear decay to $1.28 \times 10^{-5}$.

| Params | Values |
|---|---|
| Layers | 26 |
| Hidden size | 3072 |
| FFN size | 8192 |
| Vocab size | 100,288 |
| Heads | 24 |
| Key-value heads | 8 |
| Adam $\beta$ | (0.9, 0.95) |
| LR | $3.2 \times 10^{-4}$ |
| Batch size | 4M |
| Warmup steps | 1000 |
| Weight decay | 0.1 |

Table 7: Hyperparamters used for the YOCO-3B model in Section 5.1.

## E   Hyperparameters for Scaling Curves (Section 5.2)

Table 8 reports the hidden dimension, number of layers, and number of heads used for different model sizes. The head dimension of gated retention is set to 256. To align the number of parameters, the FFN size for Transformer is $\frac{8}{3}d$ while the FFN size for YOCO is $3d$. The training length is set to 2048. The batch size is set to 0.25M tokens. We use the AdamW [26] optimizer with $\beta_1 = 0.9, \beta_2 = 0.98$. The learning rate is $1.5 \times 10^{-4}$ for 160M to 1.4B sizes and $7.5 \times 10^{-5}$ for 2.7B to 13B sizes. The

warmup step is 375 with linear rate decay. The weight decay is set to 0.05. We train the models with 40k steps, i.e., 10B tokens.

| Size | Hidden Dim. | #Layers | #Heads |
|------|------------|---------|--------|
| 160M | 768 | 12 | 12 |
| 400M | 1024 | 24 | 16 |
| 830M | 1536 | 24 | 12 |
| 1.4B | 2048 | 24 | 16 |
| 2.7B | 2560 | 32 | 20 |
| 6.8B | 4096 | 32 | 32 |
| 13B | 5120 | 40 | 40 |

Table 8: Model size and hyper-parameters used for scaling curves in Section 5.2.

# F Hyperparameters for Length Extension

We progressively extend the context length to 1M tokens in Section 5.3. The length schedule is 64K, 256K, and 1M. We up-sample the documents that are longer than the training length [9]. Table 9 shows that we use different RoPE $\theta$ and learning rate for each stage.

| Training Length | 65,536 | 262,144 | 1,048,576 |
|-----------------|--------|---------|-----------|
| Learning Rate | $8 \times 10^{-5}$ | $4 \times 10^{-5}$ | $2 \times 10^{-5}$ |
| RoPE $\theta$ | 640K | 5M | 80M |
| Training Tokens | 6B | 4B | 1.5B |

Table 9: Hyperparamters used for length extension in Section 5.3.

