# OpenReview forum: "You Only Cache Once: Decoder-Decoder Architectures for Language Models"
_NeurIPS.cc/2024/Conference — NeurIPS 2024 oral_

### Official Review · Reviewer_hQmr · 2024-07-11

**Soundness:** 3
**Presentation:** 3
**Contribution:** 3
**Rating:** 7
**Confidence:** 5

**Summary:**

This paper proposes YOCO, a hybrid model that combines gated linear attention with standard attention (SA). The model stacks efficient self attention (ESA) in the first $L/2$ layers, succeeded by another $L/2$ cross-attention layers.
Notably, the output of the last ESA is shared across subsequent CA layers, thereby achieving significant parameter reduction and enabling exceptional key-value (KV) cache compression, critical for optimizing inference.

Two ESA variants are evaluated: sliding window attention and a novel gated retention method, which incorporates data-driven head-wise decay over retention.
Upon scaling YOCO to a 3-billion-parameter model trained on a corpus of 1 trillion tokens, the authors report superior performance relative to Llama-like architectures in language modeling tasks.
They also conduct some analysis on long-seq evals and observe near-perfect performance on needle-in-haystack tests and other benchmarks like Qasper.

**Strengths:**

1. YOCO's hybrid structure delivers remarkable results in needle-in-haystack scenarios and demonstrates robust performance on retrieval-centric tasks, marking a pioneering achievement.
2. The proposed data-dependent gated-retention brings great improvement against retention.
3. By facilitating substantial KV cache compression relative to standard attention, YOCO exhibits superior retrieval capabilities compared to existing linear attention models. I'm very glad to see the results of YOCO scaling to larger sizes.

**Weaknesses:**

I see no obvious disadvantages of this paper; however, the manuscript would benefit from:

1) The authors should add more comparions with  exisiting linear-time / hybrid models trained on trillions of tokens, e.g., RWKV6 and TransNormer, whose checkpoints are publicly available.
2) Despite concurrent works, I suggest the authors to add discussions with Samba [1] and Mamba2 [2] in their next version.

[1] Samba: Simple Hybrid State Space Models for Efficient Unlimited Context Language Modeling

[2] Transformers are SSMs: Generalized Models and Efficient Algorithms Through Structured State Space Duality

**Questions:**

1. Some notations are confusing: 1) Regarding Eq.7, the usage of $\beta_{iB}$ suggests an accumulation effect from preceding chunks, which may mislead readers. Additionally, the notations of $\beta_{[i]}(j,k)$ appears unused. If I understand correctly, $x_{[i]}$ is a 2-d tensor while $\beta_{[i]}$ is a scalar, it could be better to use another notation to distinguish the two. 2) Eq. 8 should be $\mathrm{head}_1,\dots,\mathrm{head}_n=\dots$
2. I'm curious if the authors have tried other linear attention variants instead of gRet, e.g., Mamba, and GLA.

---

> ### Author Rebuttal · Authors · 2024-08-07
>
> Thanks for your positive comments.
>
> >Q1: add results of exisiting linear-time / hybrid models trained on trillions of tokens, e.g., RWKV6 and TransNormer
>
> A1: We focus on the evaluation using the same training data for fair comparisons. So we use OpenLLaMA and StableLM in Table 3. We also compare with other linear-time / hybrid architectures (e.g., Mamba, and hybrid H3) in Table 5 with the same training data and settings. We can include RWKV6 and TransNormer numbers for reference results as suggested.
>
>
> ---
> >Q2: Despite concurrent works, I suggest the authors to add discussions with Samba and Mamba2 in their next version.
>
> A2: Samba and Mamba2 were released in arXiv after the NeurIPS submission. These two methods are complementary to this work, which is promising to use them as self-decoder in YOCO. Specifically, the ablation setting `Interleaved & Hybrid` in Table 6 is similar to the hybrid design of Samba, and both Mamba2 and gRetNet follow the similar design principles. We can include the discussions in the camera-ready version.
>
>
> ---
> >Q3: suggestions about notations
>
> A3: Thanks for the suggestion. We will optimize the notations in Eq. (7) and (8).
>
>
> ---
> >Q4: I'm curious if the authors have tried other linear attention variants instead of gRet, e.g., Mamba, and GLA.
>
> A4: We conducted experiments with sliding-window attention and gated retention in the work. The other linear attention variants are supposed to behave similarly and follow the same trend.

---

> > ### Comment · Reviewer_hQmr · 2024-08-07
> >
> > Thanks for replying to my question, I have no other questions and keep my score.

---

### Official Review · Reviewer_qC7V · 2024-07-12

**Soundness:** 3
**Presentation:** 3
**Contribution:** 3
**Rating:** 6
**Confidence:** 4

**Summary:**

The authors propose a new architecture for language models, where the top half of the transformer layers uses the KV from the bottom layer, while the bottom half applies efficient self-attention. The proposed architecture effectively reduces the KV cache size while maintaining the performance of the model, especially for long-context scenarios. Experiments also show that the method could scale up to 13B parameters.

**Strengths:**

1. The proposed architecture is simple and effective, which could be easily integrated into existing transformer model implementations.
2. The experiments are comprehensive and convincing. The authors prove the effectiveness of the method on a 3B model and up to 1M context.

**Weaknesses:**

1. As opposed to the first strength, the paper does not introduce new techniques or insights, thus limited in novelty. The authors also did not give possible explanations for the effectiveness of the proposed architecture.
2. The paper is lack of sufficient argumentation surrounding the design decisions. Though section 4.5 and 4.6 provide some preliminary analysis, further discussions are required to make the paper more convincing. For example, how the efficient self-attention and decoder-decoder structure affect the model's performance respectively.
3. The paper reports that the model outperforms the baseline transformers, but it remains unclear what contributes to the performance improvement. The main experiment is a partial comparison of the 1T token checkpoint instead of the fully trained model, so it is possible that the model is just easy to optimize (under large learning rates) but not necessarily converge to a better point. Also, the YOCO model has a different hyperparameter setting from the baseline model, with larger intermediate size (the scaling curve), which may also contribute to the performance improvement.

**Questions:**

Please refer to **Weaknesses**

**Limitations:**

Please refer to **Weaknesses**

---

> ### Author Rebuttal · Authors · 2024-08-07
>
> >Q1: insights and explanations for the effectiveness of the proposed architecture
>
> A1: The key insights are summarized as follows. First, KV cache can be shared across layers without significantly affecting language modeling performance. Most previous work focuses on compressing KV cache along with the sequence dimension, while YOCO improves the cache issues from another perspective, i.e., the layers. Second, the hybrid design is competitive. After the NeurIPS submission, there were several concurrent works indicate this insight, such as Samba, and character.ai's architecture blog. Third, the early exit insight (as described in Figure 2 and Line 115) dramatically improves the prefill speed. All the above insights and advantages make YOCO novel and go beyond conventional KV compression methods, which improves deployment and user experience.
>
>
> ---
> >Q2: How the efficient self-attention and decoder-decoder structure affect the model's performance respectively.
>
> A2:
>
> - The comparisons between decoder-decoder and decoder-only architectures are presented in Table 6, i.e., the settings `YOCO_[1:1]` and `Interleaved & Hybrid`, where the interleaved model is a decoder-only architecture with hybrid layers. The results show that the two layouts achieve similar performance.
>
> - For different self-decoder choices, we conducted experiments with sliding-window attention and gated retention. Both representative design choices work well as shown in Figure 3 (i.e., model size scaling up experiments) and Table 5 (i.e., ablation studies).
>
> - Different ratios between self-decoder and cross-decoder are also compared in Table 6.
>
> - In order to comprehensively inspect how the proposed architecture affects performance, we conducted evaluation from diverse perspectives, including scale up training tokens (Section 4.1), scaling curves of the proposed architectures (Section 4.2),  scale up the YOCO model to 1M context length and evaluate its long-sequence modeling capability (Section 4.3), compare with Transformer variants (Section 4.5), and ablation studies on various design choices (Section 4.6).
>
>
> ---
> >Q3: what contributes to the performance improvement?
>
> A3: The improvements mainly come from the hybrid-style architecture. Multiple recent works confirmed this point, such as Samba[1], and Jamba[2]. The trends are consistent across different learning rate schedules. Because YOCO saves the key and value projection matrices, for fair comparisons, we accordingly increase the FFN part in order to keep the overall parameter count similar across models. For example, as shown in the page 47 of the Llama 2 paper[3], this is a common practice for fair comparisons across design choices. Besides, instead of performance, we focus more on the improvements in terms of inference memory, prefill latency, and throughput.
>
> [1] Samba: Simple Hybrid State Space Models for Efficient Unlimited Context Language Modeling
>
> [2] Jamba: A Hybrid Transformer-Mamba Language Model
>
> [3] Llama 2: Open Foundation and Fine-Tuned Chat Models
>
>
> We hope the above explanation clarifies the rationale behind our experiment designs. Thank you again for the valuable feedback.

---

> > ### Comment · Reviewer_qC7V · 2024-08-09
> >
> > I appreciate the authors for replying to my questions. Considering the impressive performance and hybrid architecture of this work, I would like to increase the final rating to 6.

---

### Official Review · Reviewer_fhCA · 2024-07-13

**Soundness:** 3
**Presentation:** 3
**Contribution:** 4
**Rating:** 7
**Confidence:** 4

**Summary:**

The paper introduces YOCO, a decoder-decoder architecture designed for large language models. This architecture comprises a cross-decoder stacked upon a self-decoder, efficiently encoding global key-value caches reused by the cross-decoder. YOCO aims to reduce GPU memory demands and improve prefill latency and throughput while maintaining global attention capabilities. Experimental results demonstrate that YOCO achieves competitive performance compared to Transformer models, significantly reducing inference memory and prefill latency, and effectively extending context lengths up to 1M tokens with high retrieval accuracy.

**Strengths:**

- YOCO's design, with its cross-decoder and self-decoder, offers a novel approach to caching key-value pairs, reducing GPU memory consumption.
- The architecture significantly reduces prefill latency and improves throughput, addressing critical bottlenecks in long-sequence language model inference.
- YOCO demonstrates effective scalability in model size and training tokens, maintaining competitive performance with other leading Transformer models.
- Extensive experiments validate YOCO's performance and efficiency gains, showing substantial improvements in memory usage and latency across various model sizes and context lengths.

**Weaknesses:**

- Transformers with flash attention could also scale to 1m tokens (e.g. FlashDecoding, https://crfm.stanford.edu/2023/10/12/flashdecoding.html) any comparison/discussion? Additional complexity with the cross-decoder and self-decoder mechanisms may pose implementation challenges.

- While the architecture shows significant improvements in inference efficiency involving very long context lengths, it remains unclear how the fixed-size sliding window size affects the performance versus efficiency tradeoffs.

**Questions:**

- The evaluation primarily focuses on memory and latency improvements. Does YOCO also bring training efficiency gains?

- Are YOCO models slower than models in  Table 3? Since the context size is usually much smaller, but YOCO used fixed window size of 1024 while most task examples probably contain <1024 tokens.

**Limitations:**

The paper does not explicitly discuss any limitations.

---

> ### Author Rebuttal · Authors · 2024-08-06
>
> >Q1: The comparison and discussion with FlashDecoding.
>
> A1: Flash-Decoding and kernel fusion have been used in comparison (described in L240, L121, L25), i.e., the Transformer results have been based on FlashDecoding. The contributions of YOCO and FlashDecoding are orthogonal. YOCO optimizes the pre-filling complexity and KV cache memory from the perspective of architecture design, while FlashDecoding optimizes the implementation. We can directly utilize FlashDecoding for cross-decoder without rewriting the kernel.
>
>
> ---
> >Q2: The performance versus efficiency tradeoffs of Efficient Self-Attention (ESA)
>
> A2: Table 3/4 and Figure 3/4 show that ESA does not harm the end-to-end performance under the YOCO architecture. We find that the window size from 1024 to 4096 of self-decoder (SWA) achieves similar end performance in our early experiments.
>
>
> ---
> >Q3: The training efficiency in YOCO
>
> A3: The training efficiency of YOCO and Transformer is comparable when the training length is small. When the training length becomes long, YOCO training is faster compared with Transformers because of the cost saving of self-decoder, with a speedup ratio between 1x and 2x.
>
>
> ---
> >Q4: The efficiency comparison when the token length is very short
>
> A4: Even for short sequences, there is still 2 times prefill speedup with YOCO. As described in Figure 2 and Line 115, we can still exit early before entering the cross-decoder during the prefill stage. The YOCO models are not slower than Transformers in Table 3.

---

> > ### Comment · Reviewer_fhCA · 2024-08-11
> >
> > I thank the authors for the clarifications. I will keep my rating.

---

### Official Review · Reviewer_CYFA · 2024-07-15

**Soundness:** 4
**Presentation:** 4
**Contribution:** 4
**Rating:** 8
**Confidence:** 4

**Summary:**

The paper introduces YOCO (You Only Cache Once), a novel decoder-decoder architecture for large language models. YOCO uses a self-decoder to generate global key-value (KV) caches, reused by a cross-decoder, reducing GPU memory usage and improving inference efficiency. The architecture achieves comparable performance to full transformers but with significantly lower memory demands. Extensive experiments show YOCO's effectiveness in scaling with more training tokens, larger model sizes, and longer context lengths, up to 1 million tokens. YOCO demonstrates substantial improvements in memory footprint, prefill latency, and throughput, making it a promising model for long-context understanding and multimodal applications.

**Strengths:**

Overall, this is a high-quality paper.

Originality: The paper presents a novel architecture that achieves performance comparable to full transformers with only one layer storing global KV tokens.

Quality: The paper includes extensive experiments that robustly demonstrate the proposed model structure's ability to maintain excellent scaling performance while achieving good inference efficiency. The experiments are comprehensive and well support the claims made in the paper.

Clarity: The paper is well-motivated, clearly stating the problem it aims to solve. The overall model structure is also clearly explained. The experimental section is well-organized, effectively showcasing how the model scales up with more training tokens, larger model sizes, and longer context lengths. It was very enjoyable to read.

Significance: I believe this paper highlights the importance of achieving good scaling performance with only a single layer of global KV cache, including strong needle retrieval capabilities. This is a significant contribution, demonstrating the potential for efficiently handling long sequences with such models.

**Weaknesses:**

- The paper should evaluate the in-context learning ability of the new architecture.

- I believe more ablation studies on the window size of the sliding-window attention are necessary. The paper could more thoroughly investigate several important model parameters.

- I think a significant future application for long context models is long video understanding. While this paper focuses on language modeling, it could benefit from including some discussion on extending the model to multimodal scenarios.

- There are a few typos in the paper. For example, in line 36, "early exit before entering the self-decoder" should be "cross-decoder" instead of "cross-encoder."

**Questions:**

- In the ablation study, does Unstacked YOCO refer to the model without the self-decoder?

- Therefore, in the new model, will the number of layers and the number of attention heads per layer differ from the standard transformer design?

**Limitations:**

Please refer to the weakness section.

---

> ### Author Rebuttal · Authors · 2024-08-06
>
> Thank you for the positive review and insightful feedback.
>
> >Q1: In the ablation study, does Unstacked YOCO refer to the model without the self-decoder?
>
> A1: The input of Unstacked YOCO's cross-decoder is the output of **embedding layer**. In comparison, the input of YOCO's cross-decoder is the output of **self-decoder**. The model without the self-decoder is `YOCO_[0,1]` in Table 6, where the whole model is stacked with cross-decoder and the shared KV cache is word embedding.
>
>
> ---
> >Q2: The model hyper-parameters such as the number of layers and the number of attention heads.
>
> A2: We keep most of them the same as the standard Transformer to ensure fair evaluation, where both the model size and depth are comparable.
>
>
> ---
> >Q3: a significant future application for long context models is long video understanding
>
> A3: We consider multimodal scenario as one of the most important future directions. Thanks for your suggestions.
>
>
> ---
> >Q4: Ablation studies on the window size of SWA.
>
> A4: We find that the window size from 1024 to 4096 achieves similar end performance in our early experiments. Since a larger window size affects inference latency, we keep the default as 1024.
>
>
> ---
> >Q5: a few typos
>
> A5: We will fix them in the camera-ready version.

---

> > ### Comment · Reviewer_CYFA · 2024-08-11
> > **Thanks for the rebuttal**
> >
> > Thanks for addressing most of my concerns, I will keep my score.

---

### Decision · Program_Chairs · 2024-09-25

**Decision:**

Accept (oral)

**Comment:**

This paper proposes a novel decoder-only architecture that uses only one layer global KV cache to improve inference efficiency. Reviewers all agree this is a highly novel paper with clear writing and thorough experiments (similar performance with vanilla transformer, high memory efficiency, low latency, and scaling to 1m context). Overall this is a great contribution to the community.